# Fog Computing Model to Orchestrate the Consumption and Production of Energy in Microgrids

**DOI:** 10.3390/s19112642

**Published:** 2019-06-11

**Authors:** Eric Bernardes C. Barros, Dionísio Machado L. Filho, Bruno Guazzelli Batista, Bruno Tardiole Kuehne, Maycon Leone M. Peixoto

**Affiliations:** 1Computer Department, Federal University of Bahia (UFBA), Salvador 40170110, Brazil; eric.bernardes@ufba.br (E.B.C.B.); maycon.leone@ufba.br (M.L.M.P.); 2Computer Department, Federal University of MS (UFMS), Ponta Pora 79907414, Brazil; dionisio.leite@ufms.br; 3Computer Department, Federal University of Itajuba (UNIFEI), Itajuba 37500903, Brazil; brunoguazzelli@unifei.edu.br (B.G.B.); brunokuehne@unifei.edu.br (B.T.K.)

**Keywords:** smart grid, microgrid, fog, cloud, energy distribution model, power grid, performance evaluation

## Abstract

Energy advancement and innovation have generated several challenges for large modernized cities, such as the increase in energy demand, causing the appearance of the small power grid with a local source of supply, called the Microgrid. A Microgrid operates either connected to the national centralized power grid or singly, as a power island mode. Microgrids address these challenges using sensing technologies and Fog-Cloudcomputing infrastructures for building smart electrical grids. A smart Microgrid can be used to minimize the power demand problem, but this solution needs to be implemented correctly so as not to increase the amount of data being generated. Thus, this paper proposes the use of Fog computing to help control power demand and manage power production by eliminating the high volume of data being passed to the Cloud and decreasing the requests’ response time. The GridLab-d simulator was used to create a Microgrid, where it is possible to exchange information between consumers and generators. Thus, to understand the potential of the Fog in this scenario, a performance evaluation is performed to verify how factors such as residence number, optimization algorithms, appliance shifting, and energy sources may influence the response time and resource usage.

## 1. Introduction

The growing increase in energy demand complemented by the decrease in the means of energy production point to the need for a new energy infrastructure. Researchers have considered the use of smart grids, demand-side management, and distributed generation as solutions to this problem, where the energy generation and consumption aligned with sensors that collect data from the grid can provide cost savings and pollutant reduction [1,2].

Several services need to be created to handle all data being sent from the various smart grid layers, such as generators, consumers, and the power transmission means, for this new infrastructure to work [2]. A Cloud with services available to treat each smart grid layer can be used.

One of the main goals of smart grids is to reduce energy costs, improve user satisfaction, and control the peak hours’ distribution [3]. Thus, these services must have algorithms that generate information about which equipment can be turned on or off during peak hours. To achieve these goals, this type of decision must be made with the help of demand-side response techniques, where several authors have studied how integer linear programming, genetics algorithms, and various optimization algorithms can help reduce peak hours and increase consumer comfort [1,3,4,5,6,7,8,9,10,11,12].

In addition, if we consider energy production, we can verify how much the energy generated by the wind or the photo-voltaic panels can supply the demand and then decrease the nonrenewable energy production. Therefore, these algorithms will receive data from all household appliances and need to treat each home according to their consumption policies besides monitoring the energy production from different sources; this means that a huge amount of data will be generated and passed across the network.

If only one house with simple appliances attached to one measurement unit (PMUor Smart Meter) is considered, 30–120 samples are generated per second [13], since each house has several appliances that need to be monitored and controlled. If we consider four houses, we have four-times more data being transmitted and processed per second. This data processing is necessary to perform the demand response management helping the consumer to save on the energy bill, but for this, it is necessary to choose the equipment quickly that can be connected and to know the type of energy source that is being used. If renewable energy sources are being used, it is necessary to make fast choices, since this type of production is not constant and depends on the environment. These applications need to have a strict latency requirement between 100 ms and 5 s [14].

The International Data Corporation (IDC) expects the device number with sensors to increase from 50 billion–1 trillion in 2020 [15]. The increase in the number of Internet of Things (IoT) and sensing devices has led to an exponential growth of the data produced, while computational resources such as bandwidth and available memory may not be sufficient. Currently, Cloud Computingis being used as a way to offer agility, availability, and scalability, significantly increasing the system reliability. However, this centralized infrastructure model may not be appropriate for emergency cases that require quick responses with low latency and low jitter. Another Cloud-related problem is the requirement of sending all collected data for analysis in Cloud data centers, increasing the communication cost on the link, as well as increased storage and processing costs. One way to minimize costs in data transfer is to perform the partial processing of services at the edge of communication through Fog Computing, smoothing the problem of the growth of data production directly at its source. Thus, Fog Computing, as an extension of Cloud Computing, is used to perform services directly at the edge of the network, providing low latency and real-time computing. In this case, the Fog can provide infrastructure to store and process data for real-time decisions [16,17,18].

Thus, the Fog can be used within this new infrastructure in two ways: **first**, to reduce the communication data traffic exchange, and **second**, to move from a conventional smart grid into a Microgrid. The idea is to take advantage of the concept of distributed generation that comes with using small renewable energy producers and to allow a Microgrid to operate either connected to the national centralized power grid or in power island mode. Therefore, Fog computing can obtain data on the amount of energy produced and consumed in real time to perform processing tasks and reduce the need for Cloud operators and still process information with high bandwidth and low latency, and with the help of the Cloud, it is possible to make decisions that reduce the energy bill and the amount of pollutants [4].

Authors such as Shahryari [6] and Zahoor [4] have already proposed the use of the Fog to assist the energy management of homes and industries. Jamil [5], Wang [7], and Muralitharan [8] carried out studies on the use of optimization algorithms to move appliances away from peak hours. Yet, none of the related works used optimization algorithms within the Fog or Cloud and performed energy production control. They also did not perform a performance evaluation to understand how some factors such as residence number, energy sources, optimization algorithms, and household appliance shifting impact the Fog or Cloud performance environment.

Therefore, this paper proposes the use of Fog computing to optimize the use of electric energy in Microgrids. This optimization is obtained in two ways. The first is in energy consumption, where the demand response is applied to shift household appliances to schedules where there is no peak energy consumption and thus decreases bill. The second is in energy production where a PID (Proportional Integral Derivative) controller is used to balance the renewable production, non-renewable production, and consumption. We use optimization algorithms and a PID controller running in the Fog layer to orchestrate the energy resources. Due to the need to centralize the information obtained from the consumption and production of energy, we provide an analysis of how the fog servers behave when they are simultaneously performing the calculations of the controller PID and the appliance shifting. Finally, the performance evaluation is done showing the use of genetic algorithms and an ordinary FIFO algorithm to verify the factors that most impact the Fog in each scenario.

This paper is organized as follows: Section 2 provides the concepts about the smart grids. Section 3 presents the related works and their relation to the proposed work. Section 4 describes the evaluation and results obtained. Section 5 presents the conclusions.

## 2. Smart Grids and Fog Computing

Smart grids are designed to save energy through information communication technology, where it is possible to obtain a fully-observable power distribution network, where both the utility and consumer interact with each other for information sharing [4]. Yet, to achieve these properties, it is necessary to perform consumption control and power generation [19]. Thus, it is necessary to collect data and interact with the smart grid topology, obtaining real-time communication from consumers, distributors, transmission lines, and generators; see our architecture model in Figure 1.

This model where data are collected from sensors and sent to servers has been used for several solutions such as: car traffic analysis, security cameras, temperature control. All of these services are important for the development of smart cities, but they need to send data over the Internet for real-time information, which can lead to network saturation and the demand for more computing power. Therefore, new architectures need to identify and transmit only relevant data and thus reduce the transmission of data over the network.

Thus, using Fog computing can be useful for fast data processing along with low-latency communication. The problem with this approach is that the Fog servers do not have great computing power. Therefore, the proposed architectures also need the help of the Cloud, where high storage and processing power are possible. Within the scope of smart grids, this processing is important because in order to save energy and control the energy life cycle, robust algorithms need to be executed quickly.

These algorithms receive data from energy production and consumption. Through these data, it is possible to detect the peak hours’ consumption and moments of high renewable production and then perform the calculations for shifting consumption.

### 2.1. Peak Control

Peak hours occur when most people use appliances at the same time, so the price of electricity tends to increase, since power generators need to increase production by spending more resources per hour. The demand response tries to control the peak hours by shifting household consumption to times when there are no peak hours [1].

The problem with this approach is reducing the bill without losing comfort since you cannot use an appliance when you want. Thus, it is necessary to understand the functioning of the residences in order to reduce the bill and increase the comfort of the consumer.

### 2.2. Scheduling Appliances

To reduce the bill and increase the comfort, shifting policies can be made by the consumer. However, these policies need to be optimized by a central module to get all the necessary information in real time and use optimization algorithms to choose when the rate is less expensive and without creating a new peak time.

The scheduling problem for an appliance is formulated as an optimization problem that can be solved in terms of electricity cost, peak hours, and waiting time [1]. Thus, the algorithms can be executed using these three parameters to find the best solution of the optimization problems.

### 2.3. Optimization Algorithms

Optimization problems are solved by maximizing and minimizing functions of one or more variables in a given domain. The algorithms used to solve these problems can be either deterministic or probabilistic.

The scheduling of appliances done by shifting consumption to non-peak hours can be implemented with optimization algorithms belonging to the probabilistic class, such as evolutionary algorithms. This approach is used because a large number of combinations may be necessary to find the best time to connect the equipment [8].

Genetic algorithms are one type of evolutionary algorithm and are used to search for optimal solutions to different problems. Therefore, we can use this class of algorithms to look for the best solution for home appliance shifting based on consumer policy and renewable energy production.

### 2.4. Communication between Layers

Smarts grids were created so that there is information exchange between their layers, something that does not exist in the conventional grids. This exchange of information makes possible real-time decisions that help in the intelligent use of energy.

Cloud and Fog computing can help this communication if used within an architecture that allows orchestration between robust Cloud processing and the speed of the Fog computing situated at the edge of the network [4].

### 2.5. Renewable Production

Unlike the consumption peak hours, which spends much resource to produce energy, renewable energy production can receive the peak hours of energy production. If it is considered that renewable resources such as the Sun and wind are unstable, production by these means can achieve a large energy production, which can help in non-renewable production, reducing pollution and the energy bill. Therefore, it is necessary to develop mechanisms that control the process of energy production considering the inconstancy of the alternative production [12].

### 2.6. PID Controller

The PID controller is an method widely used in industry to perform process control. This control is done by monitoring the process variable by sensors that send the data to the control system. These values are compared to the set point, and depending on the desired value, the system activates the actuators so that the value approaches the desired value.

To keep the Process Variable (PV) close to the Set Point (SP), the system needs to manipulate the Manipulated Variable (MV), which is the variable that the controller acts on to control the process, such as the amount of fuel that will be burned to generate energy or the voltage applied to a heating resistor. Thus, the control increases or decreases MV to produce more or less energy according to necessity; this variation is reached through Equation (1).
(1)$$MV\left(t\right)=Kp\left[E\left(t\right)+\int E\left(t\right)dt+Kd\frac{dE(t)}{dt}\right]$$

In this paper, the PID controller was developed in Fog computing to monitor and act on the production of renewable and non-renewable energy. Thus, the amount of non-renewable energy produced will depend directly on the renewable energy being produced at the time. Therefore, the SP will be the production of renewable energy less consumption as in Equation (2). The PV needs to approach the SP, and for this, the proportional, integral and derivative techniques are used to find out how much MV needs to be changed.
(2)$$SP=\sum_{i=1}^{t_{rp}}rp_{t}-\sum_{j=1}^{t_{c}}C_{j}$$
where $rp_{i}$ is the renewable energy being produced at the moment and $C_{j}$ is the houses’ consumption at that moment.

Therefore, the variation of MV is directly linked to the consumption or the production of renewable energy, and as MV is responsible for the fuel that will be spent it is important to use optimization techniques in energy consumption and to improve the use of renewable energy.

## 3. Related Work

Some researches have evaluated the best algorithms that can be used to ensure the demand response using smart grid concepts. However, none have performed tests considering the information in real time using Fog computing as a way to manage the entire energy life cycle for the best results, besides considering the PID controller to balance the load between the generators. However, several studies point to important results such as the use of meta-heuristic algorithms in house management, as Nadeem et al. (2018) [1], where a control for home energy management based on meta-heuristic algorithms such as Teaching-Based Optimization (TLBO), Genetic Algorithm (GA), Firefly Algorithm (FA), and Optimal Stopping Rule (OSR) theory was proposed. The main objective of this proposal was to reduce energy consumption and maximize consumer comfort. For this, the best features of the existing algorithms were combined, and three hybrid algorithms were created: OSR-TLBO, OSR-GA, and OSR-FA; tests were performed in MATLAB to verify the performance of these algorithms. The authors also suggested the use of Fog or Cloud computing as future work to achieve gains in scalability. Thus, the behavior of the Fog and Cloud in the proposed algorithms was not studied.

Sedighizadeh et al. (2019) [9] introduced a new stochastic optimization method for short-term scheduling of energy and reserve in Microgrids considering energy storage constraints. The Differential Evolutionary (DE) and Modified PSO (MPSO) algorithms were used considering the incentive based on the demand response program and generation reserve scheduling. The tests were performed in MATLAB, and the results showed that applying the algorithm to the energy management model, it was possible to reduce the operating cost by 7.8% and the emission of pollutants by 18.03% if compared to the MINLP method.

Javaid et al. (2018) [3] employed the load shifting strategy, to decrease total electricity payment; for this was proposed a hybrid algorithm that used the techniques of the bat and crow algorithm. In this study, simulations were performed in a single house with 15 appliances that used the Critical Peak price (CPP) scheme to compare the bat algorithm, the crow algorithm, and the hybrid algorithm. The results were obtained in tests carried out in MATLAB and showed that load was successfully shifted to lower price time slots, which ultimately led to a 31,191% reduction in total electricity payment.

Jamil et al. (2018) [5] proposed a Home Energy Management System (HEMS) in which appliance scheduling was performed using the load shifting strategy. Algorithms such as Cuckoo search, earthworm optimization, and a hybrid technique cuckoo-earthworm optimization were used for scheduling the smart home appliances. The hybrid technique HCEOdecreased the electricity cost by 49% with respect to unscheduled electricity cost because it scheduled most of the appliances during the low price hours.

Muralitharan et al. (2016) [8] implemented the Multi-Objective Evolutionary Algorithm (MOEA) to reduce energy costs and minimize the waiting time of the execution of appliances. The limitation of traditional evolutionary algorithms handling only a single objective or a weighted combination was explained, and it could not change the values of the parameters simultaneously. The results showed that the proposed method minimized the cost of electricity and the waiting time of the appliances for costumers.

Logenthiran et al. (2012) [10] presented a demand-side management strategy based on a load shifting technique. The day-ahead load shifting technique proposed was mathematically formulated as a minimization problem where a heuristic-based Evolutionary Algorithm (EA) that easily adapted heuristics in the problem was developed for solving this minimization problem. Simulations were carried out on a smart grid, which contained a variety of loads in three service areas, one with residential customers, another with commercial customers, and the third one with industrial customers. The simulation results showed that the proposed demand-side management strategy achieved substantial savings, while reducing the peak load demand of the smart grid with gains of 5% in residential areas, 5.8% in commercial areas, and 10% in industrial areas.

Wan et al. (2018) [11] proposed an Energy-aware Load Balancing and Scheduling (ELBS) method based on Fog computing. This proposal was validated and compared with the Central-Station Control scheduling (CSCS) method in experiments at a candy packaging manufacturer, and the experimental results showed that the proposed method provided optimal scheduling and load balancing for the mixing work robots. Comparisons between the Fog and Cloud were also performed, and the results showed that the information obtained from the Fog nodes was more accurate.

Izadbakhsh et al. (2015) [12] implemented the scheduling of energy sources within a microgridcomposed of micro-turbines, photovoltaics, fuel cells, and wind turbines through a framework that dealt with the reduction of operating costs and emissions. This framework used the Normal Boundary Intersection (NBI) technique to solve the multi-objective problem and generate the Pareto set and the fuzzy method for making decisions.

Wang et al. (2018) [7] established an optimization model for economical operation of a microgrid. This operational model was implemented using a GA, and a better operational strategy was determined by the clean energy resources and the demand response program. In order to test these models, an intelligent park microgrid consisting of photovoltaic power generation, a combined cooling and power system, an energy storage system, and response load was used. Finally, it was observed that this model could effectively reduce the operation cost of and improve the utilization rate of renewable energy sources.

Zahoor et al. (2018) [4] proposed a three-layered architecture for smart buildings where a Fog infrastructure was developed at the network edge to collect private data through smart meters. A performance analysis of a Cloud-based system was performed considering the Fog as an intermittent layer, showing that the use of the Fog can be efficient at reducing the network bottleneck.

Shahryari et al. (2017) [6] introduced the dynamic scheduling of appliances with the help of the Cloud and Fog. This approach suggested the use of a demand-side queue of priorities where consumers were prioritized by their importance, their consumption policies, and the status of energy resources.

Unlike these works, we propose the Fog Computing Paradigmas an extension of Cloud Computing. In our research, Fog Computing performs services directly at the edge of the network, providing low latency and real-time computing, essential in cases where urgent flow changes are required. In addition, these papers did not cover performance evaluation aspects in their analyses; see Table 1.

## 4. Performance Evaluation

Performance Evaluationis an important technique that evaluates any system, allowing one to verify the accuracy, validity, and meaning of the magnitude produced during the evaluation. Furthermore, it is used to obtain the highest statistical accuracy as possible, providing the maximum information with a minimum number of experiments, showing the effects of various factors on the observed result. In this paper, performance evaluation was used to select appropriate optimization algorithms and in which environment they can be better deployed (Cloud or Fog).

### 4.1. Scenario

In this paper, the model used for evaluation considers the data that are obtained from energy generation and houses’ consumption. Thus, renewable energy sources such as solar and wind and non-renewable energy sources such as diesel turbines will be considered. The consumption side will have up to four houses with seven appliances (refrigerator, freezer, washing machine, dishwasher, TV, light, and microwave), but only the first four can carry out the shifting. The washing machine and the dishwasher can only be turned on when the energy consumption is not during the peak time. The refrigerator and freezer cannot turn on ice production at peak times. The rest of the appliances can be connected according to the will of the consumer.

The energy price rate varies according to the renewable energy production and the consumption peaks. Thus, the greater the renewable energy production, the lower the energy bill of the consumer. This choice was made to encourage the use of renewable energy, which reduces the pollutants’ emission.

The algorithms created will be tested in two scenarios: the first in the Fog, which will process the First In First Out (FIFO) algorithm, where it is checked if the devices can be turn on at that moment; otherwise, they will enter the shift queue and will turn on during the first time the energy is less expensive; the second in the Cloud will process a Genetic Algorithm (GA) that will check the conditions of the previous day to decide the best time to turn the equipment on the next day. The goal is to see if a Fog can manage a Microgrid with real-time data and compare the results with the processing in the Cloud. In energy production, the PID controller is used in both scenarios; see Figure 2.

### 4.2. Genetic Algorithm Implementation

In the genetic algorithm implementation, we consider that each residence has a maximum value that can be spent per hour and that each appliance has a consumption and a time to finish its task. Thus, the appliances’ consumption being used at that time cannot exceed the maximum value; see Equation (3). The genetic algorithm searches for an appliance combination that does not exceed the consumption limit per hour. A table with the devices’ consumptions used in GA can be seen in Table 2. As the consumption price depends on the energy type being produced (renewable or non-renewable), the maximum and minimum value that each appliance can spend per hour is given in Table 3.
(3)$$MC_{m}\geq \sum_{i=1}^{T_{a}}AC_{i}*T_{m}$$
where $MC_{m}$ is the maximum consumption of the moment, AC appliance consumption (kWh) and $T_{m}$ the tariff at that moment. The tariffs used were: 0.15 for renewable energy and 0.45 for non-renewable energy. These values mean that the tariff varied between 0.15 and 0.45, this variation depending on which type of energy was most used, and a proportional value was applied.

At the beginning of the algorithm execution, a random population with two possible solutions was created. This population was a set of chromosomes that had Genes 0 and 1. These genes represent the appliances, and the value of one will be considered for appliances that can be turned on, otherwise zero. Each chromosome will be calculated and evaluated so that it does not exceed the maximum limit of that moment. If the maximum limit is exceeded, the grade of this chromosome will be one (worst evaluation).

After the evaluation, the chromosomes passed through the crossover and mutation operators. In the crossover, these chromosomes will be randomly mixed. These mixtures will be the new solutions. In the mutation, each gene will have a 0.01 probability of being inverted. After these changes, new solutions will be compared with the old ones, and those that have a better evaluation will remain. The final solution will be given at the end of 15 iterations.

### 4.3. Experiment Planning

The experiments performed on the power grid layer were performed using the GridLab-d simulator. GridLab-d is a simulation tool that allows the study of several problems belonging to smart grids, such as demand response and renewable integration [20]. Thus, GridLab-d was used to create homes, appliances, and power generators, as well as sending consumption and power generation data across the network to the Fog or Cloud. Each home could consume up to 550 kWh/month; the diesel turbine could produce up to 5000 kWh/month; and renewable (solar and wind) power production could produce up to 2500 kWh/month.

Our Fog consisted of a Raspberry PI 3 that had a 1.2-GHz Quad Core CPU and 1 GB of RAM. The server Apache Tomcat 9 was installed on the Raspberry, which received the data coming from GridLab-d and executed the algorithm FIFO, as well as the calculation of the control PID. The Cloud was a virtual machine located in gcloud that had a 3.5-GHz Intel Xeon with 4GB of RAM with the same installation of the Fog, the only difference being the scheduling algorithm used was the GA. Considering this infrastructure, the cloudwas 42.07% slower than the fog, but within the range of 100 ms–5 s in the experiments performed.

In order to evaluate the performance of the algorithms, full factorial experiment planning $2^{k}$ was used, presented in [21], where two levels were defined for each factor *k*. A full factorial design (Table 4) measures the response variables by using all the possible combinations among the factors levels. Our design consisted of four factors (A, B, C, and D), which all having two levels. It can be expressed as a full factorial $2^{4}$ design. Thus, we used the experiment model according to Equation (4).
(4)$$\begin{array}{c} Y_{ijkw}=\mu +A_{i}+B_{j}+C_{k}+D_{w}+AB_{ij}+AC_{ik}+AD_{iw}+\\ \;BC_{jk}+BD_{jw}+CD_{kw}+ABC_{ijk}+ABD_{ijw}+ACD_{ikw}+\\ \;BCD_{jkw}+ABCD_{ijkw}+\epsilon_{ijkw} \end{array}$$

This μ represents the overall mean effect given by the replication requirement of 10 times (α=0.05) with all seeds well tested. We also considered the error term (ϵ) in the model. Besides, we assumed in this work the linear regression model, which is the relationship between the dependent variable $y_{i}$ and the linear regression vector $x_{i}$. The general model through the error can be obtained by $y_{i}=\beta_{1}x_{i1}+\cdots +\beta_{p}x_{ip}+\varepsilon_{i}=x_{i}^{T}\beta +\varepsilon_{i},$ for i=1,…,n, where ${}^{T}$ denotes the transpose, so that $x_{i}^{T}\beta$ is the inner product between the vectors xi and β. Often, these *n* equations are put together and written in vector form as **y** = **X**β + ε, where: $$y=\left(\begin{array}{c} y_{1}\\ y_{2}\\ \vdots \\ y_{n} \end{array}\right),\;$$
$$X=\left(\begin{array}{c} x_{1}^{T}\\ x_{2}^{T}\\ \vdots \\ x_{n}^{T} \end{array}\right)=\left(\begin{array}{ccc} x_{11} & \cdots & x_{1p}\\ x_{21} & \cdots & x_{2p}\\ \vdots & \ddots & \vdots \\ x_{n1} & \cdots & x_{np} \end{array}\right),$$
$$\beta =\left(\begin{array}{c} \beta_{1}\\ \beta_{2}\\ \vdots \\ \beta_{p} \end{array}\right),\;\varepsilon =\left(\begin{array}{c} \varepsilon_{1}\\ \varepsilon_{2}\\ \vdots \\ \varepsilon_{n} \end{array}\right).$$

The experiments were performed according to Table 5, and the results of CPU usage and request times can be seen in Figure 3 and Figure 4.

### 4.4. Results’ Analysis

Regarding energy savings, houses that used the FIFO algorithm reduced their energy bills by between 4% and 6% in relation to homes that did not use any kind of demand response. Houses that used the GA algorithm had between a 5% and 8% reduction in their energy bills. This savings was made possible by the shift in energy consumption, which shifted the consumption of some household appliances to times when renewable energy production was greater than the total consumption of the houses; see Figure 5 and Figure 6.

As can be seen in Figure 5, the change in renewable energy was accompanied by the PID controller that used the total energy consumption and the variation of the renewable energy to control the gas turbine production. Energy production had some problems at a time when power consumption was at its highest, between 18 and 22 h, because at that time, solar production decreased, leaving only wind production.

An example of the energy consumption of houses can be seen in Figure 6, where Houses 01 and 04 were using the demand response with the FIFO and GA algorithms, respectively, and Houses 02 and 03 were not participating in the demand response. As can be seen, houses that used the demand response, when they perceived that the production of renewable energy decreased, also reduced their energy consumption by turning off some appliances and shifting that consumption to the moment that the renewable production was greater.

In addition to energy saving data, we verified the main factors responsible for calculations, which can be seen in Figure 7; the shifting was the factor that most influenced both the CPU (57.78%) and the request time (38.49%). This was due to the amount of equipment that needed to be analyzed; whenever there was no appliance shifting (Experiments 2, 6, 10, and 14), the amount of processing decreased (see Figure 8). The same assertion can be made regarding time, the only difference being that no-shifting requests occurred on the Cloud, in which the time of the request tended to increase due to Internet traffic (Figure 9).

The second major influence factor was the generation of energy (17.83%). This was because whenever the renewable energy was considered, the PID controller needed to perform the calculation of the control variable, which increased the CPU time and, consequently, the time of the requests (13.67%).

## 5. Conclusions

The concern of this paper was with the peaks of energy and the conditions of adjustment between sources of renewable and non-renewable energy in a Microgrid environment. Currently, most of these adjustments are made on the Cloud. However, some peaks of energy are seasonal and reactive, turning Cloud Computing into a non-viable solution because of its non-desirable network latency, even after a peak of energy or between them. Hence, we used Fog Computing with low latency to perform the electric energy control in a Microgrid.

The Fog Computing proposed in this paper was an extension of the cloud, in which was made available management services such as the PID controller and scheduling algorithms for appliances in order to reduce consumers’ bills. Different from the related works, the scheduling algorithms executed in our Fog used real-time data from the GridLab-d simulator. These algorithms (FIFO and GA) used the PID calculations to know the current energy rate and then performed the scheduling, performing the appliance shifting to hours when the energy was less expensive.

The request time reduction reached in this paper allowed saving energy and managing the use of non-renewable energies. Our approach showed that the Fog Computing infrastructure can help with algorithm execution time by shortening the time of each request by up to 21.49% in relation to the cloud, providing low latency and real-time computing.

Performance Evaluation showed through the experiments that it was possible to reduce the energy bills. The Fog demonstrated that it can perform the calculations required to manage a Microgrid, providing faster returns and maintaining a CPU utilization rate below 50% during requests.

## Figures and Tables

**Figure 1 sensors-19-02642-f001:**
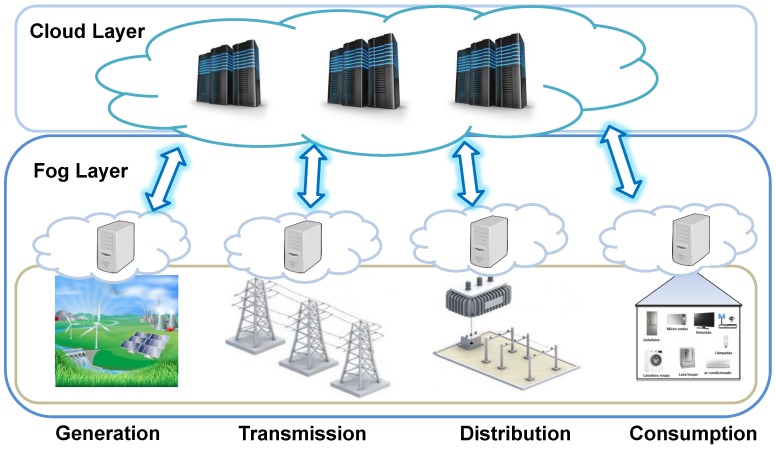
Smart grid topology aided by the Fog and Cloud.

**Figure 2 sensors-19-02642-f002:**
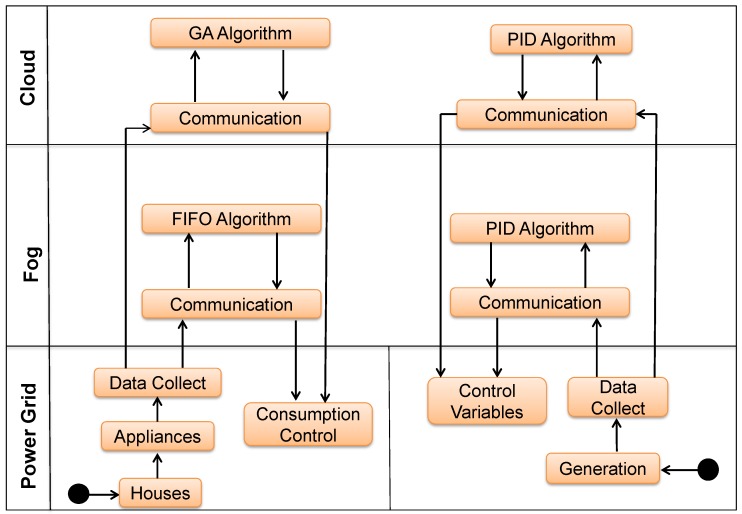
Power generation managed by the PID controller and GA and FIFO algorithm in the Fog.

**Figure 3 sensors-19-02642-f003:**
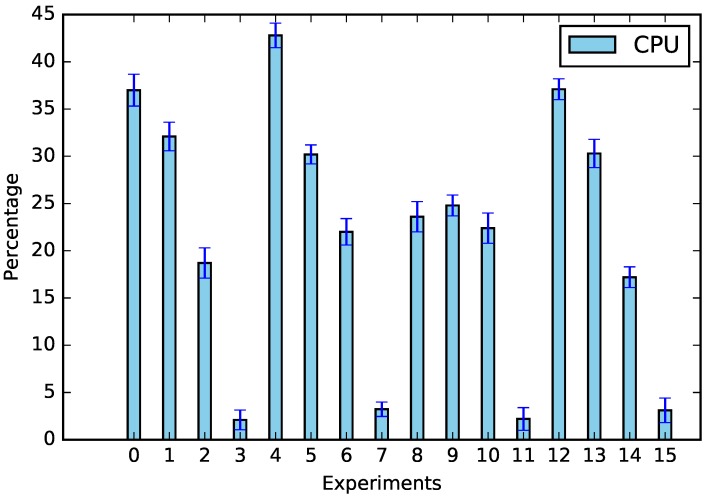
CPU usage. The experiments can be seen in Table 5.

**Figure 4 sensors-19-02642-f004:**
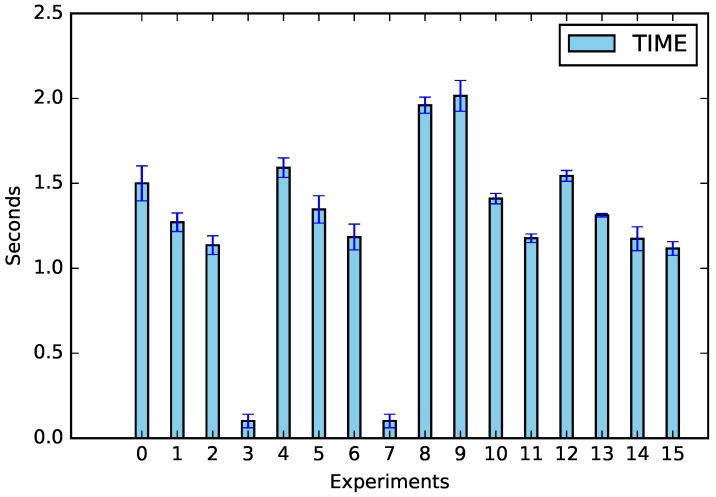
Waiting time for the request.

**Figure 5 sensors-19-02642-f005:**
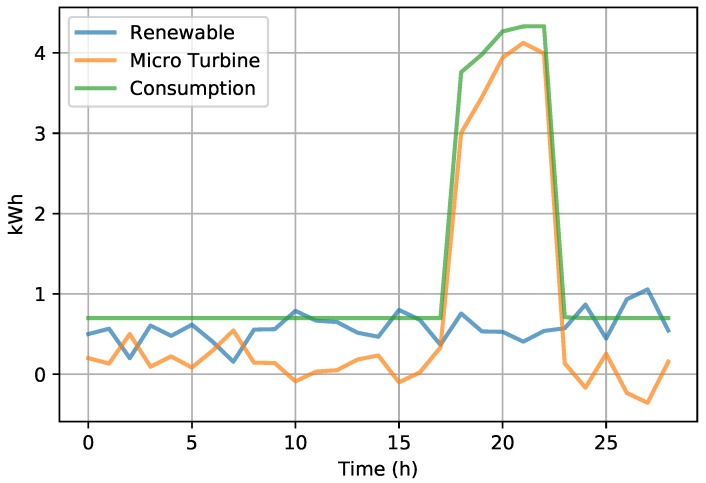
Average power generation managed by the PID controller in the Fog.

**Figure 6 sensors-19-02642-f006:**
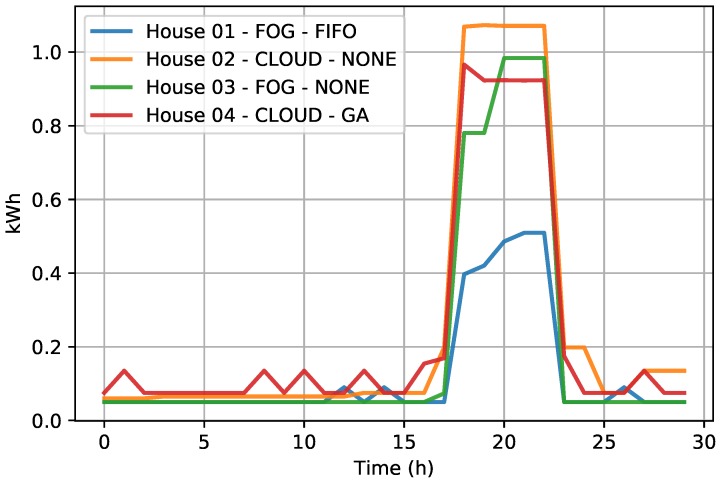
Comparison of power management algorithms and infrastructures.

**Figure 7 sensors-19-02642-f007:**
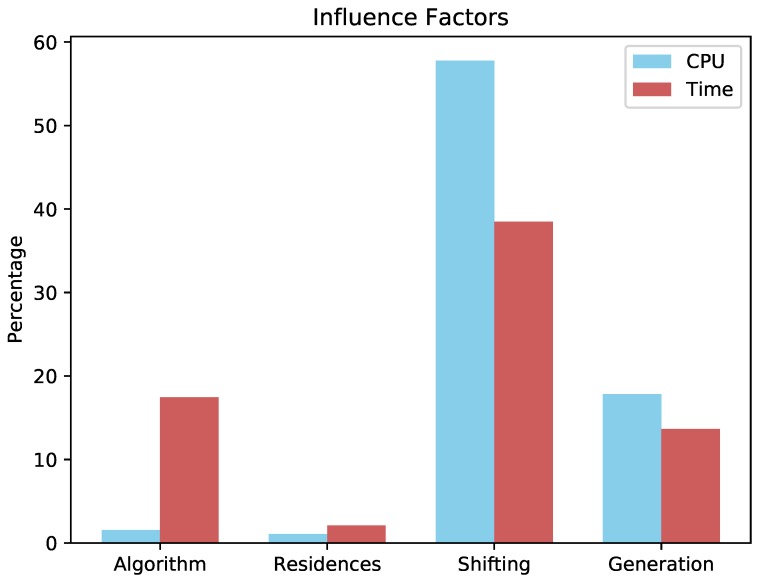
Influence factors in the studied scenario.

**Figure 8 sensors-19-02642-f008:**
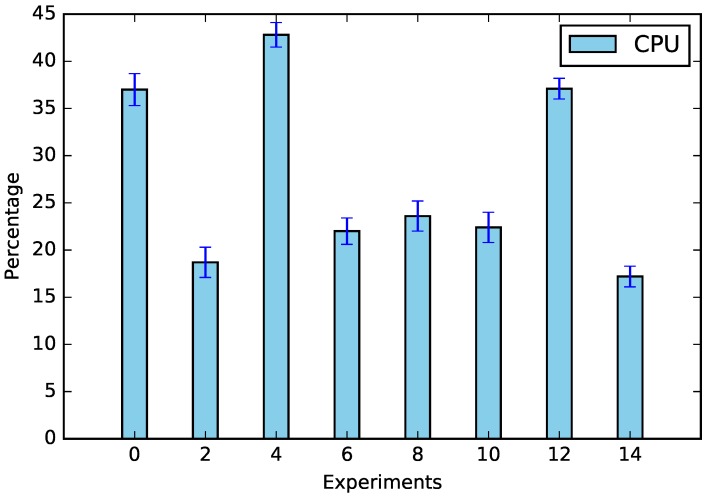
CPU usage with and without the appliance shifting.

**Figure 9 sensors-19-02642-f009:**
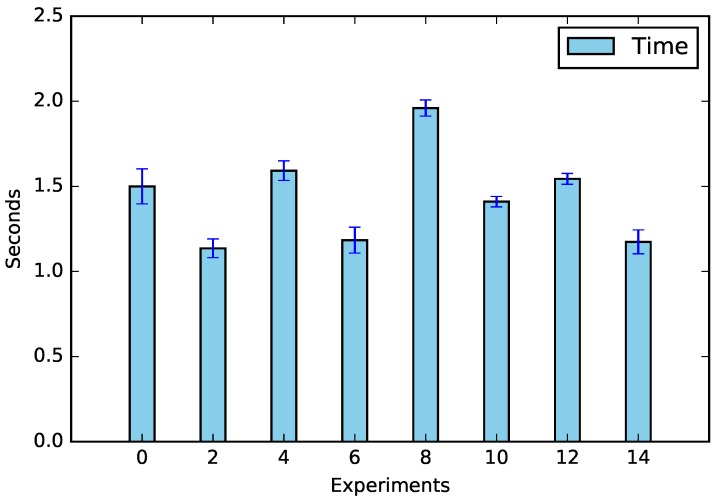
Time usage with and without the appliance shifting.

**Table 1 sensors-19-02642-t001:** Related work.

Authors	Peak Control	Fog	Cloud	Optimization	Scheduling	Clean Energy	Communication Layer	PDI Control
(Shahryari, 2017)		√	√		√		√	
(Wan, 2018)		√	√	√	√		√	
(Zahoor, 2018)		√	√	√			√	
(Nadeem et al., 2018)	√			√	√			
(Logenthiran et al., 2012)	√			√				
(Sedighizadeh et al., 2019)	√			√		√		
(Izadbakhsh et al., 2015)				√		√		
(Jamil, 2018)	√			√	√			
(Wang et al., 2018)	√			√	√	√		
(Muralitharan, 2016)	√			√	√			
(Javaid, 2018)	√			√	√			
**(This paper)**	√	√	√	√	√	√	√	√

**Table 2 sensors-19-02642-t002:** Consumption.

Appliance	kWh
Refrigerator	2
Freezer	1.5
Washing Machine	0.47
Dishwasher	1.2

**Table 3 sensors-19-02642-t003:** Price.

Appliance	Energy Type	Price/kWh
Refrigerator Minimum	RNW	0.3
Refrigerator Maximum	N-RNW	0.9
Freezer Minimum	RNW	0.225
Freezer Maximum	N-RNW	0.675
Washing Machine Minimum	RNW	0.0705
Washing Machine Maximum	N-RNW	0.2115
Dishwasher Minimum	RNW	0.18
Dishwasher Maximum	N-RNW	0.54

**Table 4 sensors-19-02642-t004:** Factors and levels.

Factors	Levels	
Algorithm	*First in First out*	*Genetic Algorithm*
Appliance	Shifting	Non-shifting
Residences	2	4
PID Controller	Renewable	Diesel turbine

**Table 5 sensors-19-02642-t005:** Experiments.

Exp	Algorithm	Residences	Appliance	PID Controller
0	FIFO	2	SHIFT	RNW
1	FIFO	2	SHIFT	N-RNW
2	FIFO	2	N-SHIFT	RNW
3	FIFO	2	N-SHIFT	N-RNW
4	FIFO	4	SHIFT	RNW
5	FIFO	4	SHIFT	N-RNW
6	FIFO	4	N-SHIFT	RNW
7	FIFO	4	N-SHIFT	N-RNW
8	GA	2	SHIFT	RNW
9	GA	2	SHIFT	N-RNW
10	GA	2	N-SHIFT	RNW
11	GA	2	N-SHIFT	N-RNW
12	GA	4	SHIFT	RNW
13	GA	4	SHIFT	N-RNW
14	GA	4	N-SHIFT	RNW
15	GA	4	N-SHIFT	N-RNW

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
