# Peer review of "Fog Computing Model to Orchestrate the Consumption and Production of Energy in Microgrids"

_sensors, 2019, doi:10.3390/s19112642_

Round 1

Reviewer 1 Report

The paper describes algorithms applied to optimized energy usage and production in smart grids, and particularly Microgrid. A distinctive feature of the work is the use of Fog computing to mitigate network latency and provide prompt decisions regarding appliance usage considering energy production (renewable, non-renewable), peak hours and energy price.

Major issues

While this is a hot topic and more research is needed, I am not satisfied with the way the whole idea is presented. A major issue is that Fog computing is proposed, on the grounds of lower latency at the expense of less powerful resources. The practical reasons driving this necessity are not discussed at all. For example, what are common data volumes/generation rate in this domain? (this should be presented upfront, e.g. in the Introduction). The same applies to the paragraph starting with “Thus, using Fog computing can be useful for fast data ...” (page 3, opening paragraph). Why latency is so harmful to these kind of systems/applications?

In page 2, what are these “factors”? Since the paper claims to be modeling/tackling the problem in a different way compared to existing works, these should be exhaustively presented and clearly discussed.

I understand that Fog resources might be less powerful than Cloud resources, but a more distinctive aspect is the network latency to reach them. Thus, the network architecture assumed in the approach should be better characterized in the Introduction or opening Sections. The same applies to page 3, “problem with this approach is that the Fog servers do not have a great computing power, ...”. This explanation is missing. By the way, using a Pi versus a more powerful server in the experiments does not mean one has a Fog/Cloud layered system. The experiments do not take network latency into account at all.

In the introduction, the paragraph “A Microgrid operates either connected to the national centralized power grid or singly, as a power island mode. Microgrid address these challenges using sensing technologies and large-scale computing infrastructures for building smart electrical grid.” is confusing. Here, “these challenges” (disconnection in my view) does not naturally lead to using large-scale computing infrastructures, but rather the need for smart energy demand control.

Later in the paper, “Due to the lower CPU capacity (IoT devices), we provide a full understand report of how Fog servers behave when they are using this type of algorithm.” This is vague. Which IoT devices? Aren't sensing devices considered “IoT” devices only? Moreover, what is this report? Finally, “Full understand report” seems grammatically incorrect.

Page 2, right column: “the critical maximum price scheme“. What is this scheme?

The ending paragraph in Section 2 should be heavily revised/extended. Isn't using PID control and even controlling energy production other important contributions of the work? By the way, it is not clear whether the idea is to innovate by optimizing both demand side and production sides, and if this is actually an innovative aspect of the paper w.r.t. existing works.

The paper lacks examples illustrating the various concepts, mechanisms and motivations. In line with my previous comments about showing real data representing common data generation rates, demands, prices, and so on, how is this “shifting” done in practice (Section III.A)? What are the constraints therein?

The paper fails to explain why is actually important to ensure that “robust algorithms need to be executed quickly” (Section III). While this is intuitive, the impact of latency in bills, energy produced, and other relevant variables is not quantified experimentally. I would had liked to see experiments showing how different latencies affect the optimization capability of each algorithm. This is a huge gap in the paper since it claims to provide a performance evaluation as a distinctive feature of the work w.r.t. works published in the literature.

Section III.B, reference [13]: It is unclear why this is treated as a tri-objetive problem, since it seems that electricity cost and peak hours are highly correlated variables. As a first sigh, it should be a two-objetive problem instead.

It is surprising that the paper jumps directly from the assumptions made and warm-up explanations of the approach (Section III) to its evaluation (IV). The actual “meat”, such as how the problem is formulated via GAs, chromosome representation, used operators, and many more, are not explained.

Section IV.A: This scenario, while realistic, seems too simplistic and synthetic. Why not employing real data? I wonder whether the authors made an effort to find/adapt a real dataset for the simulations. The same applies to the numbers shown in the first paragraph of Section IV.B.

In table III: Shouldn't “PID algorithm” be another level of row “Algorithm”?

Minor issues

Page 1: “where several authors study”. Please provide citations.

Page 1: “trafficked”. Is this the right word here?

Page 2: “PID control” -> PID controller?

Page 2. Section 2 presents the proposed work, Section 3 -> . Section 3

Opening paragraph, Section 2: “... perform the demand response, but none have performed tests considering information in real time using Fog computing as the manager the entire energy life cycle for best results and ...“. Please check grammar.

Around fig 1: Why the paper says “all layers”? Just two layers are depicted. Do you mean “all steps” (there are four of these)?

Page 3: “these attributes” -> “these properties”?

Fig 2 caption is misleading: GA and FIFO levels are used in addition to PID control.

Author Response

1) While this is a hot topic and more research is needed, I am not satisfied with the way the whole idea is presented. A major issue is that Fog computing is proposed, on the grounds of lower latency at the expense of less powerful resources. The practical reasons driving this necessity are not discussed at all. For example, what are common data volumes/generation rate in this domain? (this should be presented upfront, e.g. in the Introduction). The same applies to the paragraph starting with “Thus, using Fog computing can be useful for fast data ...” (page 3, opening paragraph). Why latency is so harmful to these kind of systems/applications?

Thank you for pointing it. We fixed it in the new paper version. Informations considering the amount of data generated in production (solar, wind and gas turbines) and energy consumption (from home appliances) was included in the introduction. When is considered only one measurement unit (PMU or Smart Meter), it can be produced about 30 to 120 samples per second1. As the environment increases, more data are produced. These information data are related to constantly monitoring of energy consumption and production. It was also included in the paper the reason for low latency requirement in Microgrids. Briefly explaining here, in a Microgrid is demanded a strict latency ranging of 100ms to 5 seconds2. To address these requirements of fast communication infrastructure that can deal with huge amount of data in such real-time environment, we use Fog Computing paradigm. Fog extends Cloud to the edge of the network, offering high-throughput, wireless access, low latency, and real-time application.

1 - P. Kansal and A. Bose, “Smart grid communication requirements for the high voltage power system,” in Proc. IEEE PES General Meeting, Jul. 2011, pp. 1–6.

2 - P. Kansal and A. Bose, "Bandwidth and Latency Requirements for Smart Transmission Grid Applications," in IEEE Transactions on Smart Grid, vol. 3, no. 3, pp. 1344-1352, Sept. 2012. doi: 10.1109/TSG.2012.2197229

2) In page 2, what are these “factors”? Since the paper claims to be modeling/tackling the problem in a different way compared to existing works, these should be exhaustively presented and clearly discussed.

Thanks for pointing it. We modified the related works section in order to explain that. In the context of Performance Evaluation field, a factor in a experiment is a variable that affects the response variables and can take on several levels. Primary factors can have a major impact on a response variable and should be considered, but secondary factors have a non-significant impact on the response variable. We have included factors in our experiment such as residences numbers, optimization algorithms, appliances shifting and energy sources, explaining the importance and how previous research considers the use of these factors.

3) I understand that Fog resources might be less powerful than Cloud resources, but a more distinctive aspect is the network latency to reach them. Thus, the network architecture assumed in the approach should be better characterized in the Introduction or opening Sections. The same applies to page 3, “problem with this approach is that the Fog servers do not have a great computing power, ...”. This explanation is missing. By the way, using a Pi versus a more powerful server in the experiments does not mean one has a Fog/Cloud layered system. The experiments do not take network latency into account at all.

Sorry about that and thanks for pointing it. We forgot to write in the paper that the server with Intel Xeon 3.5GHz with 4Gb of RAM is a virtual machine located in the google cloud. We corrected in the new version showing the location of the servers, the raspberry on the network edge and the Intel 3.5GHz server in cloud.  In this way, we run the experiments considering the latency of the internet.  We also added in section 4.2 that within this architecture cloud requests were 42.07% slower than the requests for fog.

4) In the introduction, the paragraph “A Microgrid operates either connected to the national centralized power grid or singly, as a power island mode. Microgrid address these challenges using sensing technologies and large-scale computing infrastructures for building smart electrical grid.” is confusing. Here, “these challenges” (disconnection in my view) does not naturally lead to using large-scale computing infrastructures, but rather the need for smart energy demand control.

Sorry about that and thanks for pointing it. Maybe we did not provide a clear and full explanation. We already fixed it in the Introduction and abstract. When was said “large-scale computing infrastructures”, we are addressing Fog-Cloud to help the control power demand and manage power production. Even when Microgrid operates as a power island mode (disconnected to centralized power grid), we are not addressing the power grid, but the network communication which remains connected through 5G, satellite or other network. However, the reviewer is also correct about the need for smart energy demand control. Therefore, we use some methods running in the Fog layer to orchestrate the energy resources.

5)  Later in the paper, “Due to the lower CPU capacity (IoT devices), we provide a full understand report of how Fog servers behave when they are using this type of algorithm.” This is vague. Which IoT devices? Aren't sensing devices considered “IoT” devices only? Moreover, what is this report? Finally, “Full understand report” seems grammatically incorrect.

Sorry about that and thanks for pointing it. We were referring to smart meters and how they are not able to centralize the information of all microgrid residences and energy production data in a single smart meter and why the use of a fog server should be used. This paragraph was rewritten.

6) Page 2, right column: “the critical maximum price scheme“. What is this scheme?

Sorry about that and thanks for pointing it. On paper we wrote critical maximum price, but in fact we wanted to have written critical peak pricing which is a pricing scheme used to calculate the cost of electricity. This scheme has maximum and minimum values that vary depending on the time of use and the real time price.

7) The ending paragraph in Section 2 should be heavily revised/extended. Isn't using PID control and even controlling energy production other important contributions of the work? By the way, it is not clear whether the idea is to innovate by optimizing both demand side and production sides, and if this is actually an innovative aspect of the paper w.r.t. existing works.

Sorry about that and thanks for pointing it. We are looking for surveys that contain power generation that use PID controller and power consumption optimization at the same time and we will add in related works. The idea of the paper is to propose a change of energy bill tariff based on the type of energy source being used in energy production, the PID controller is used to balance as needed the production of renewable and non-renewable energy , how much more renewable energy is used, cheaper the bill will be.

8) The paper lacks examples illustrating the various concepts, mechanisms and motivations. In line with my previous comments about showing real data representing common data generation rates, demands, prices, and so on, how is this “shifting” done in practice (Section III.A)? What are the constraints therein?

Sorry about that and thanks for pointing it. We are improving the new version of the paper to better illustrate the concepts and motivations involved. We are adding tables of prices of consumption by appliances and how the price variation occurs when the use of renewable and non-renewable energy occurs.

9) The paper fails to explain why is actually important to ensure that “robust algorithms need to be executed quickly” (Section III). While this is intuitive, the impact of latency in bills, energy produced, and other relevant variables is not quantified experimentally. I would had liked to see experiments showing how different latencies affect the optimization capability of each algorithm. This is a huge gap in the paper since it claims to provide a performance evaluation as a distinctive feature of the work w.r.t. works published in the literature.

Sorry about that and thanks for pointing it. We will improve these explanations by adding more examples of how the tests were performed. We perform the tests with the purpose of trying to reduce the latency through the use of a fog infrastructure. The question is: if we have little processing, but a simpler algorithm will we be able to decrease the latency without affecting the energy bill? In other words, is a simpler algorithm at a lower latency capable of decreasing the energy bill more than a more robust algorithm, but which is at a higher latency? The answer was that in relation to houses that do not have any type of demand response, we achieved a reduction between 4% and 6% using the FIFO. While using GA we achieved despite a higher latency a result between 5% and 8%. We then performed a performance assessment to see how the processing and response time of these case studies behaved.

10) Section III.B, reference [13]: It is unclear why this is treated as a tri-objetive problem, since it seems that electricity cost and peak hours are highly correlated variables. As a first sigh, it should be a two-objetive problem instead.

Sorry about that and thanks for pointing it. The waiting time of each appliance only has an impact on consumer comfort. When we mention this goal, we are considering the use of a consumer policy that could interfere with the algorithms, giving priority to some equipment due to the consumer's will.

11) It is surprising that the paper jumps directly from the assumptions made and warm-up explanations of the approach (Section III) to its evaluation (IV). The actual “meat”, such as how the problem is formulated via GAs, chromosome representation, used operators, and many more, are not explained.

Sorry about that and thanks for pointing it. We created a new subsection in section 4 to describe how the genetic algorithm was developed in experiments. We consider how much renewable energy is being produced and how much nonrenewable energy is being produced, with renewable energy in the experiments being cheaper. Each customer has a value that he can spend per hour. Thus, considering the consumption and the time that each equipment needs to finish its task, thus the genetic algorithm is applied to choose the cheapest combination of consumption for that moment.

12) Section IV.A: This scenario, while realistic, seems too simplistic and synthetic. Why not employing real data? I wonder whether the authors made an effort to find/adapt a real dataset for the simulations. The same applies to the numbers shown in the first paragraph of Section IV.B.

We appreciate the reviewer's comment. We tried to create a real simulation environment where it was possible to control the appliances by the network. Gridlab-d was the only tool we found that allowed this control. We did the experiments using raspberry where it is possible to connect appliances inside the gridlab-d or real appliances. The problem with the second approach is to get the resources needed to equip homes with these appliances. Thus, the use of the network is real, but the generation of production and energy consumption is done by the simulator.

13) In table III: Shouldn't “PID algorithm” be another level of row “Algorithm”?

Sorry about that and thanks for pointing it. The PID controller is being considered a factor with 2 levels: renewable energy and non-renewable energy. In the text, we call this factor energy sources, but we will switch in the new version for PID controller. There are two variables that can affect the PID controller, the residential consumption and renewable energy. When we consider the level where the non-renewable energy is operating, the only variation that is occurring is the consumption. When we consider renewable energy the PID is being affected by consumption and the renewable energy production, which may change depending on the environment.

Minor issues

Page 1: “where several authors study”. Please provide citations.

Page 1: “trafficked”. Is this the right word here?

Page 2: “PID control” -> PID controller?

Page 2. Section 2 presents the proposed work, Section 3 -> . Section 3

Opening paragraph, Section 2: “... perform the demand response, but none have performed tests considering information in real time using Fog computing as the manager the entire energy life cycle for best results and ...“. Please check grammar.

Around fig 1: Why the paper says “all layers”? Just two layers are depicted. Do you mean “all steps” (there are four of these)?

Page 3: “these attributes” -> “these properties”?

Fig 2 caption is misleading: GA and FIFO levels are used in addition to PID control.

Sorry about that and thanks for pointing it. We fixed all the recommendations.

Reviewer 2 Report

The contribution of this paper is considered relevant and novel in the field of the smart grid and the trend in moving t computing to the edge as much as possible. Results, for the proposed scenario, are convincing. Nevertheless, to assess the real contribution, the paper should be reorganized, be more descriptive and justify all the decisions to sustain the results. 

In the introduction the authors say that “the paper proposes the use of Fog Computing to optimize the use of electric energy in Microgrids”. This description of the contribution is too broad, especially, if we consider that just after this sentence, the authors mention PID control which has not been introduced in advance. In general, I recommend the authors to reorganize the introduction to provide a better presentation of the field, the problem, the motivation of the solution and the structure of the paper. This section should be self-contained and the reader shouldn’t need extra reading to understand a general perspective.

The related work is considered up-to-date but, again, this section should be reorganized to improve readability and connection among paragraphs, I would suggest to put a story line to avoid a list of related work. 

Regarding the section Smart Grid, I recommend the authors to move the general part to a section just after the introduction and before the related work (as a background); meanwhile a particular section would be devoted to the PID control as the deeper knowledge to be applied in the proposal. BTW, the meaning of PID is not included in the paper. A more extensive description of PID control should be introduces as the main support for the rest of the contribution.

The major concerns about the paper are related with the validation scenario, both the presentation and the scenario itself. 

Regarding the presentation, the decision about using FIFO and GA algorithms for Fog and Cloud respectively is not clearly justified. Be careful with the names, i.e, is controlled variables the same as manipulated variables in PID control? I recommend to put clearly separate the description of the scenario (with the justification) and the results and the evaluation of the performance. The description of the scenario should include a more detailed description of the data and the architecture of the scenario. 

Meanwhile microgrid intends solving smart grid challenges using sensing technologies andlarge-scale computing infrastructures, as it is stated by the authors, the validation scenario (implemented in a simulator) should justify  that the scale (number of nodes) is realistic for the challenges. 

The evaluation of performance is difficult to follow and should be reorganized to make the conclusions clearer and sounder. 

-  To summarize above issues, if the authors propose using a fog solution to reduce latency and data exchange, it could be unrealistic to do the experiments in a scenario with few nodes. Also, the results of the experiment shouldn’t focus on power saving but on the success in reducing latency and data exchanges. 

Author Response

1) In the introduction the authors say that “the paper proposes the use of Fog Computing to optimize the use of electric energy in Microgrids”. This description of the contribution is too broad, especially, if we consider that just after this sentence, the authors mention PID control which has not been introduced in advance. In general, I recommend the authors to reorganize the introduction to provide a better presentation of the field, the problem, the motivation of the solution and the structure of the paper. This section should be self-contained and the reader shouldn’t need extra reading to understand a general perspective.

Sorry about that and thanks for pointing it. We modified the introduction and specified that fog is being used to execute optimization algorithms that calculate the demand response and the PID controller. We also defined the concepts of demand response and PID controller in the introduction.

2) The related work is considered up-to-date but, again, this section should be reorganized to improve readability and connection among paragraphs, I would suggest to put a story line to avoid a list of related work.

Sorry about that and thanks for pointing it. We reorganized the related works section trying to make it easier to understand in the new version.

3) Regarding the section Smart Grid, I recommend the authors to move the general part to a section just after the introduction and before the related work (as a background); meanwhile a particular section would be devoted to the PID control as the deeper knowledge to be applied in the proposal. BTW, the meaning of PID is not included in the paper. A more extensive description of PID control should be introduces as the main support for the rest of the contribution.

Sorry about that and thanks for pointing it. We followed the  reviewer recommendation and changed the section on Smart Grid for after the introduction, we also added more information  about the PID  (Proportional-Integral-Derivative) controller and why it was implemented.

4)  Regarding the presentation, the decision about using FIFO and GA algorithms for Fog and Cloud respectively is not clearly justified. Be careful with the names, i.e, is controlled variables the same as manipulated variables in PID control? I recommend to put clearly separate the description of the scenario (with the justification) and the results and the evaluation of the performance. The description of the scenario should include a more detailed description of the data and the architecture of the scenario.

Sorry about that and thanks for pointing it. We perform the tests with the purpose of trying to reduce the latency through the use of a fog infrastructure. The question is: if we have little processing, but a simpler algorithm will we be able to decrease the latency without affecting the energy bill? In other words, is a simpler algorithm at a lower latency capable of decreasing the energy bill more than a more robust algorithm, but which is at a higher latency? The answer was that in relation to houses that do not have any type of demand response, we achieved a reduction between 4% and 6% using the FIFO. While using GA we achieved despite a higher latency a result between 5% and 8%. The PID controller is used to balance consumption with power generation, this controller is important because it tells you which type of energy was most used to produce power and thus change energy tariffs. The algorithms use these tariffs to verify if can turn on or off the appliances at that time.

5) Meanwhile microgrid intends solving smart grid challenges using sensing technologies and large-scale computing infrastructures, as it is stated by the authors, the validation scenario (implemented in a simulator) should justify  that the scale (number of nodes) is realistic for the challenges.

Sorry about that and thanks for pointing it. We changed the term large-scale infrastructure to Fog-Cloud infrastructure. In the experiments, the simulation was done to generate data on energy generation and energy consumption. The Gridlad-D simulator was chosen because it allowed the sending of this data through the network. The tests considered the use of a real network, where the data is sent to the cloud gcloud or to the raspberry pi. In the experiments we only got a raspberry pi, which eliminated the possibility of load distribution between servers fog. Thus, we consider a smaller number of houses with up to 7 appliances to perform processing and measure the response time of each request. f we consider combinations of these appliances we can have up to 7! combinations for each house, that gives 5040 combinations for each house and each combination can change every minute. Although the genetic algorithm has been chosen to avoid such calculations in brute force, it still needs to store historical information about previous combinations to perform the evolution and choose the best combination, which also requires a great computational effort.

6) The evaluation of performance is difficult to follow and should be reorganized to make the conclusions clearer and sounder.

Sorry about that and thanks for pointing it. We reorganized the performance evaluation section trying to make it easier to understand in the new version.

7)  To summarize above issues, if the authors propose using a fog solution to reduce latency and data exchange, it could be unrealistic to do the experiments in a scenario with few nodes. Also, the results of the experiment shouldn’t focus on power saving but on the success in reducing latency and data exchanges.

As we explained in the previous review, the simulator was used to generate data on energy production and consumption. The data is trafficked in a real network, to consider more residences in the experiments we would need more raspberry pi and perform the load balancing and thus test the latency of the network.

Round 2

Reviewer 1 Report

- Please explain what critical peek price is (page 5). In addition, "peek" -> "peak"?
- All cross-references and citations appear unresolved in the pdf file ("??").
- With respect to the authors' response "Sorry about that and thanks for pointing it. We are improving the new version of the paper to better illustrate the concepts and motivations involved. We are adding tables of prices of consumption by appliances and how the price variation occurs when the use of renewable and non-renewable energy occurs.". I could not find a table of prices (just consumption in Kwh).
- Regarding my comment #9, no new experiment was added. The generality of the claims cannot not be fully supported by the experiments reported.
- It is still not clear why the problem is treated as a tri-objetive problem, since as I observed in the earlier round of revision, it seems that electricity cost and peak hours are highly correlated variables. I understand that the third variable, which is associated to user's conform, is clearly a separate objective.

Author Response

Please explain what critical peek price is (page 5). In addition, "peek" -> "peak"?

Sorry about that and thanks for pointing it. We corrected the spelling mistake and put it in the new version. CPP (Critical Peak Price) is a dynamic price scheme, where a tariff is used to calculate the electricity cost at the moment. These values change according to the peak hours. Thus, the value of the electricity consumption at a time (t) can be given by pricing (t) * pricing signal (t), for example.

- All cross-references and citations appear unresolved in the pdf file ("??").

Sorry about that and thanks for pointing it. It must have been an error in the last version, but we have corrected the problem.

- With respect to the authors' response "Sorry about that and thanks for pointing it. We are improving the new version of the paper to better illustrate the concepts and motivations involved. We are adding tables of prices of consumption by appliances and how the price variation occurs when the use of renewable and non-renewable energy occurs.". I could not find a table of prices (just consumption in Kwh).

Sorry about that and thanks for pointing it. It must have been an error in the last version, but we have corrected the problem.

- Regarding my comment #9, no new experiment was added. The generality of the claims cannot not be fully supported by the experiments reported.

Sorry about that. This request is very interesting, but unfortunately the time is not enough. It would be necessary to search for a dataset related to the problem, perform the experiments, plot the graphs and re-discuss the results and the relationship between what is proposed and what was obtained. In the first round we were given 10 days, now in the second 7 days, this fact makes the execution requested by the reviewer unviable. We would very much like to carry out this experiment model, since it seems to be very promising. We are committed to listing this requirement in future work.

- It is still not clear why the problem is treated as a tri-objetive problem, since as I observed in the earlier round of revision, it seems that electricity cost and peak hours are highly correlated variables. I understand that the third variable, which is associated to user's conform, is clearly a separate objective.

The inclusion of comfort is used to create random events within the experiments. The purpose of this inclusion is to create real situations that can influence consumption calculations. The consumer can at any time decide to turn on some appliances which would affect the maximum value that was passed. With this, new calculations need to be performed and the appliances can be shifted to a new schedule.